# Synergistic Activity of Cefiderocol in Combination with Piperacillin-Tazobactam, Fosfomycin, Ampicillin-Sulbactam, Imipenem-Relebactam and Ceftazidime-Avibactam against Carbapenem-Resistant Gram-Negative Bacteria

**DOI:** 10.3390/antibiotics12050858

**Published:** 2023-05-05

**Authors:** Marta Palombo, Federica Bovo, Stefano Amadesi, Paolo Gaibani

**Affiliations:** 1Microbiology Unit, IRCCS Azienda Ospedaliera-University of Bologna, 40126 Bologna, Italy; 2Department of Agricultural and Food Sciences, Alma Mater Studiorum-University of Bologna, 40126 Bologna, Italy

**Keywords:** antimicrobial resistance (AMR), carbapenem-resistant bacteria, gram-negative bacteria, cefiderocol, synergy

## Abstract

Limited treatment options are among the main reasons why antimicrobial resistance has become a leading major public health problem. In particular, carbapenem-resistant *Enterobacteriales* (CRE), *Pseudomonas aeruginosa* and *Acinetobacter baumannii* have been included by the World Health Organization (WHO) among the pathogens for which new therapeutic agents are needed. The combination of antibiotics represents an effective strategy to treat multidrug-resistant (MDR) pathogen infections. In this context, the aim of this study is to evaluate the in vitro activity of cefiderocol (CFD) in combination with different antimicrobial molecules against a collection of well-characterized clinical strains, exhibiting different patterns of antimicrobial susceptibility. Clinical strains were genomically characterized using Illumina iSeq100 platform. Synergy analyses were performed by combining CFD with piperacillin-tazobactam (PIP-TAZ), fosfomycin (FOS), ampicillin-sulbactam (AMP-SULB), ceftazidime-avibactam (CAZ-AVI), meropenem-vaborbactam (MER-VAB) and imipenem-relebactam (IMI-REL). Our results demonstrated the synergistic effect of CFD in combination with FOS and CAZ-AVI against CRE and carbapenem-resistant *Acinetobacter baumannii* (CR-Ab) clinical strains owing CFD-resistant profile, while the CFD and AMP-SULB combination was effective against CR-Pa strain displaying AMP-SULB-resistant profile. Moreover, the combination of CAZ-AVI/SULB showed synergistic activity in CAZ-AVI-resistant CRE strain. In conclusion, although further analyses are needed to confirm these results, our work showed the efficacy of CFD when used for synergistic formulations.

## 1. Introduction

Multidrug-resistant (MDR) gram-negative (GN) bacteria represent a global risk for public health and a major threat worldwide. Against MDR GN pathogens, carbapenems are considered a fundamental resource representing one of the last-line reserves available for treatment of the infections sustained by these microorganisms [1]. However, the emerging diffusion of the resistance to carbapenem significantly reduces the antimicrobial armamentarium against difficult-to-treat (DTR) infections due to MDR GN bacteria and limits the antimicrobial options available for patients with multiple co-morbidities [2]. In this context, antimicrobial treatment of the infections sustained by MDR GN in critically ill patients is considered a challenge for clinicians due to the DTR resistance phenotype and to the related high mortality rate among these patients. The World Health Organization (WHO) has defined carbapenem-resistant *Enterobacterales* (CRE), *Pseudomonas aeruginosa* and *Acinetobacter baumannii* as ‘priority 1: critical’ pathogens, for which new therapeutic agents are urgently needed [3]. Accordingly, the Centers for Disease Control and Prevention (CDC) classified MDR *P. aeruginosa* as serious threats and CR *Acinetobacter* and CRE as urgent threats [4]. In particular, among carbapenem-resistant MDR pathogens, nosocomial infections caused by carbapenem-resistant *P. aeruginosa* (CR-Pa) and carbapenem-resistant *A. baumannii* (CR-Ab) are frequently associated with high mortality and morbidity among critically ill patients, thus producing a reduction of in vitro active antimicrobial molecules against these pathogens with a consequent reduction of therapeutic options [5]. Among the family of *Enterobacteriales*, *Klebsiella pneumoniae* is the most clinically relevant species, and *Klebsiella pneumoniae* carbapenemase (KPC) represents the most frequent mechanism of resistance in several countries [6,7,8,9], since it is often associated with microorganisms harbouring different antimicrobial resistance determinants, which conferred various multidrug-resistant phenotypes. In fact, most of the widely used antimicrobial molecules, such as high-dose meropenem, colistin, phosphomycin, tigecycline and aminoglycosides, show limited activity against KPC-producing *K. pneumoniae* (KPC-Kp), thus reducing available treatment options [10].

In last years, various β-lactam/β-lactamase inhibitor combinations (βL-βLICs) were approved to treat infections sustained by MDR pathogens [11]. Despite βL-βLICs exhibiting strong in vitro and in vivo activity against CR bacteria producing β-lactamases, a growing number of strains are developing resistance to the novel combined drugs [5], posing serious limitations to these molecules. In particular, in *Enterobacteriales,* resistance to ceftazidime-avibactam (CAZ-AVI) was associated to mutations in KPC enzymes, coupled with alteration of cell permeability, while resistance to meropenem-vaborbactam (MER-VAB) and imipenem-relebactam (IMI-REL) was mainly associated with loss of porin expression, increase in *bla*_KPC_ gene copies and activation of efflux pumps [5,12]. In 2019, the U.S. Food and Drug Administration (FDA) authorized the use of a novel cephalosporin, cefiderocol (CFD) [13,14], formerly S-649266, for the treatment of urinary tract infections (UTIs) and, subsequently, for the treatment of bacterial hospital-acquired pneumonia (HAP) and ventilator-associated pneumonia (VAP). In 2020, the European Medicines Agency (EMA) approved CFD in Europe for the treatment of GN bacterial infections in adults with limited treatment options [15]. This compound demonstrated in vitro high antimicrobial activity against MDR GN pathogens, especially *Enterobacterales* and GN-nonfermenters pathogens. Equally, the results of clinical trials corroborate the in vitro data and support the CFD use in DTR infections [13,14]. The CFD success lies in its peculiar structure. Specifically, the catechol, added in C3 position, works as an iron-binding siderophore. This structural modification, on the cephalosporin backbone, makes CFD able to cross the outer membrane of GN bacteria through siderophore iron uptake systems, conferring significant protection against truncation or downregulation of porins and overexpression of efflux pumps [16]. In addition, the aminothiazole ring in C7 side chain and the pyrrolidium group in C3, improve CFD stability against β-lactamases, such as in ceftazidime and cefepime, respectively [16]. Investigating the in vitro activity of CFD against CR-Ab, CR-Pa and CRE, the 95% of 1900 CR-Ab, the 97% of 1500 CR-Pa and the 97% of 1900 CRE have a minimum inhibitory concentration (MIC) ≤ 4 µg/mL [13]. Although the majority of these CR strains demonstrate susceptibility to CFD, an alarming rate of resistance has been recently reported in some cohorts (up to 50%) [17].

To overcome these limitations, drug combinations have been proposed as a valuable strategy to increase the amount of treatment options and contain the widespread emergence of CR strains [18].

Thus, the characterization of additional synergistic antibiotic combinations represents an attractive strategy to improve treatments and hinder the increase in novel resistances. With this purpose, we investigate the potentiality of CFD-based combined drugs in three of the leading CR pathogens. Herein, we evaluate the in vitro activity of CFD in combination with ampicillin-sulbactam (AMP-SULB), piperacillin-tazobactam (PIP-TAZ), IMI-REL, MER-VAB, CAZ-AVI and fosfomycin (FOS) against well-characterized CRE, CR-Ab and CR-Pa clinical strains [19]. In addition, we evaluate the activity of sulbactam (SULB) in combination with PIP-TAZ, IMI-REL, MER-VAB, CAZ-AVI or FOS.

## 2. Results

### 2.1. Bacteria Characterization

For this study, we selected six clinical strains with varying susceptibility profiles to CFD, including two strains each of *K. pneumoniae* (CRE), CR-Ab and CR-Pa. We employed DNA sequencing to characterize genomic features that could be associated with each strain’s susceptibility pattern. The genome assembly of CRE 1 (ST512) resulted in a 5,824,243 bp genome with a G + C content of 57.00%. It was composed of 598 contigs, with an N50 of 102,792 bp and a mean length of 9739 bp. We predicted a total of 6010 coding sequences in this assembly. The genome assembly of CRE 2 (ST307) produced a 5,671,368 bp genome with a G + C content of 56.88%, composed of 113 contigs with a mean length of 50,189 bp and an N50 of 207,307 bp. We identified 5605 coding sequences in this assembly. For CR-Ab 1 (ST2), the assembly resulted in a 4,044,877 bp genome with a G + C content of 38.87%. It was composed of 123 contigs with a mean length of 32,885 bp and an N50 of 143,286 bp. We predicted a total of 3989 coding sequences in this assembly. The genome assembly of CR-Ab 2 (ST2) was 3,903,935 bp in size, with a G + C content of 38.95%. It was composed of 115 contigs with a mean length of 33,947 bp and an N50 of 141,042 bp. We identified 3787 coding regions in this assembly. The assembly of CR-Pa 1 (ST298) resulted in a 6,953,441 bp genome with a G + C content of 66.07%. It was composed of 132 contigs with a mean length of 52,677 bp and an N50 of 186,291 bp. We predicted a total of 6784 coding sequences in this assembly. Finally, the assembly of CR-Pa 2 (ST235) produced a 6,847,002 bp genome with a G + C content of 65.90%. It was composed of 213 contigs with a mean length of 32,145 bp and an N50 of 159,635 bp. We identified 6777 coding regions in this assembly. The genotypic characteristics of CR strains included in this study are summarized in Table 1. Both CRE 1 and CRE 2 *K. pneumoniae* strains carried a truncated OmpK35 porin. On the other hand, while CRE 2 also carried a truncated OmpK36, CRE 1 encoded a functional porin with a GD insertion at amino acid position 135 that is correlated with increased carbapenem MIC. For *A. baumannii* isolates, both CR-Ab 1 and CR-Ab 2 harboured a wild type OprD porin and a variant III [20] CarO porin with an additional Y245F amino acid substitution. Finally, *P. aeruginosa* strains CR-Pa 1 and CR-Pa 2 shared a wild type OprF as well as an OprD C2 variant associated with carbapenem resistance [21] with an additional G426 amino acid substitution. However, CR-Pa 2 OprD also exhibited a sequence interruption at amino acid position 64, resulting in a non-functional protein.

### 2.2. Synergy Results

From May 2021 to September 2022, we selected 6 clinical strains (n = 2) CRE, (n = 2) CR-Ab and (n = 2) CR-Pa with different susceptible profiles to CFD. The genotypic characteristics of CR strains included in this study are summarized in Table 1.

As shown in Table 2, 2/6 of isolates exhibited reduced susceptibility to CFD according to EUCAST breakpoints (CFD-R). Specifically, (n = 1) CRE and (n = 1) CR-Ab obtained MIC values greater than the breakpoint (MIC > 2 mg/L), 16 mg/L and >256 mg/L, respectively. With the purpose to perform gradient diffusion strip (GDS) synergy test, the susceptible profiles to PIP-TAZ, IMI-REL, MER-VAB, CAZ-AVI, AMP-SULB and FOS were acquired for all the included strains; results are reported in Table 2. Briefly, all analysed strains (6/6) are resistant to AMP-SULB (MIC > 8 mg/L); 5/6 of isolates have MIC > 8 mg/L for PIP-TAZ and MER-VAB, resulting as resistant to them; 4/6 of isolates have MIC values of FOS and CAZ-AVI higher than the breakpoint, MIC > 32 mg/L and MIC > 8 mg/L, respectively; and 3/4 of strains are resistant to IMI/REL (MIC > 2 mg/L). Interestingly, genomic analysis showed that the CFD-R CRE strain belonged to the epidemic K. pneumoniae ST512 clone and harboured mutated *bla*_KPC,_ as compared to the CFD-S CRE strain which belonged to ST307 and carried wild type *bla*_KPC-3_ (Table 1).

The results of the synergy tests obtained for CFD with PIP-TAZ, IMI-REL, MER-VAB, CAZ-AVI, AMP-SULB, or FOS by the GDS method are presented in Table 3. Overall, no synergic interaction was observed for CFD in combination with PIP-TAZ, IMI-REL and MER-VAB against all clinical strains (6/6). More in detail, CFD/PIP-TAZ and CFD/MER-VAB interactions prove to be additive in 2/6 of isolate, whereas the remaining 4 out of 6 strains show indifferent interaction. The CFD/IMI-REL combination results as additive against 4/6 bacterial strains and indifferent on 2 of 6 isolates. CFD added with FOS exhibited in vitro synergistic activity against 2 of the analysed strains (2/6), additive interaction on 2/6 and indifferent interaction against the last 2 out of 6 isolates. Significantly, this antibiotics combination is synergic in all the CFD-R tested strains, n = 1 CRE and n = 1 CR-Ab. CFD combined with CAZ-AVI demonstrates synergistic interaction in 2/6 of strains, specifically the CFD-R CRE and the CR-Pa CFD-S1. At the same time, this combination proves to be additional on 1/6 and indifferent in 3 of 6 isolates. The CFD/AMP-SULB combination resulted as synergic only in the CR-Pa CFD-S1 strain (1/6), additive in 1 out of 6 and indifferent in the remaining 4 of 6 isolates. Of note, all the synergistic interactions were observed in CR clinical strains showing higher MIC for CFD (i.e., ≥0.5 mg/L).

Data obtained for the synergy test of SULB combined with PIP-TAZ, IMI-REL, MER-VAB, CAZ-AVI and FOS are reported in Table 4. Indifferent interaction was detected for SULB with PIP-TAZ and IMI-REL on all clinical strains analysed (6/6). FOS/SULB and MER-VAB/SULB combinations result as additive in 2 out of 6 strains and indifferent in the remaining 4/6. CAZ-AVI added with SULB demonstrate synergic outcome on the CRE 1 isolate (1/6), additive effect against 3 out of 6 strains and indifferent interaction on the remaining 2 of 6 isolate.

## 3. Discussion

The growing emergence of MDR GN bacteria represents a significant concern due to a paucity in antimicrobial alternatives [22]. In particular, during the last two decades, the increasing rate of infections due to MDR has become a global public health threat, leading to new therapeutic requirements [5,23,24]. In light of these considerations, CRE, *P. aeruginosa* and *A. baumannii* have been included by WHO among the most critical groups of MDR bacteria requiring urgent development of treatment options [3,22,25]. In recent years, new βL-βLICs have been approved in order to overcome resistance mechanisms in resistant pathogens [26], such as carbapenemases enzymes (e.g., KPCs group which is widely distributed in different countries) [24,27], lack of porin functionality and up regulation of the efflux system [12]. However, the emergence of resistance to these new agents has been recently described [5,12], presenting the significant impact in choosing between treatment options. Despite SULB having been proven to be effective against *A. baumannii,* probably owing to the binding to penicillin-binding proteins [28], CR-Ab infections are increasing due to a deficiency in therapeutic options against CR-Ab strains. Recently, CFD, a novel siderophore-conjugated cephalosporin approved in Europe in 2020, was added to the current antimicrobial armamentarium of clinicians due to its extensive activity against several GN pathogens’ major resistance mechanisms [24] for the treatment of patients with limited therapeutic options [26]. Lately, the synergistic effect against class D carbapenemase-producing carbapenem-resistant *A. baumannii* strains was demonstrated when SULB and AMP-SULB were combined with avibactam and CAZ-AVI, respectively [29]. Lim et al. demonstrated that the FOS/SULB combination could be effective for the treatment of MDR *A. baumannii* [30]. The synergistic combination of CAZ-AVI with amikacin (AMK) or aztreonam (AZT) was active in *P. aeruginosa* and *K. pneumoniae* strains, while CAZ-AVI/MEM and CAZ-AVI/FOS were valid combinations against *P. aeruginosa* and *K. pneumoniae*, respectively [22]. The in vitro activity of MER-VAB/CAZ-AVI against CAZ-AVI- and MER-VAB-resistant KPC-Kp strains was recently demonstrated [31].

Based on these findings, the aim of this study was to evaluate the in vitro impact of CFD and SULB in combination with PIP-TAZ, IMI-REL, MER-VAB, CAZ-AVI or FOS towards CRE, CR-Ab and CR-Pa clinical strains.

Our results showed that CFD presented a synergistic effect against CRE 1- and CR-Ab 1 CFD-resistant strains when FOS was combined. In particular, results showed how in the CRE CFD-resistant strain harbouring KPC variants, which led to an increase in resistance [15,32], the combination with FOS led to a synergistic action. Furthermore, the synergistic effect of the CFD/FOS combination was revealed in the CR-Ab strain with OXA variants but sensitive to FOS. In addition, our results showed that in the CFD- and CAZ-AVI-resistant CRE strains, CFD in combination with CAZ-AVI revealed synergistic effect. Moreover, CAZ-AVI/SULB synergistic activity was found in the CAZ-AVI-resistant CRE 1 strain, demonstrating that the resistance to single antibiotics can show a synergistic profile when used in combination.

In *P. aeruginosa*, resistance to β-lactam antibiotics is related to efflux pumps MexAB-OprM, MexEF-OprN and MexCD-OprJ, and resistance to carbapenems and cephalosporins is conferred by acquired β-lactamases, such as OXA and PSE [33]. CR-Pa 1 showed synergistic activity when tested with CFD in association with CAZ-AVI and AMP-SULB, despite its resistance profile.

This study presents some limitations: first, it is not completely representative as it includes a limited number of isolates; second, the number of β-lactamases combinations tested should be increased to better understand the in vitro efficacy of new antibiotic formulations.

## 4. Conclusions

In conclusion, our results revealed that CFD might be the most effective partner when used for synergistic combination. Further studies are needed to confirm these results, since the development of new antibiotic association could represent an important therapeutic support in clinical practice.

## 5. Materials and Methods

### 5.1. Bacteria Characterization and Whole-Genome Sequencing (WGS)

All clinical strains included in this study were isolated from patients recovered at S. Orsola-Malpighi Hospital over a period spanning from May 2021 to September 2022. Species identification was performed by matrix-assisted laser desorption/ionization time-of-flight (MALDI-TOF) mass spectrometry using the MALDI Biotyper system (Bruker Daltonik, Germany). Antimicrobial susceptibility testing (AST) was carried out using the automated MicroScan Walkaway-96 system (Beckman Coulter, Brea, CA, USA), and MIC values were further confirmed by MIC test strips (Liofilchem, Italy). MIC tests were performed in triplicate. The obtained MICs were interpreted following the European Committee for Antimicrobial Susceptibility Testing (EUCAST) clinical breakpoints v12.0 (available at https://www.eucast.org/clinical_breakpoints/, accessed on 1 January 2023). Carbapenemase production was evaluated by the NG-Test CARBA 5 (NG Biotech, France) platform and confirmed with molecular assay (Xpert Carba-R, Cepheid). DNA sequencing was carried out in order to characterize the genomic features of the isolates included in this study and to investigate a potential involvement in the susceptibility profile. Genomic analysis was performed as previously described [28]. Briefly, the DNeasy Blood&Tissue Kit (Qiagen, Basel, Switzerland) was employed to extract the genomic DNA from pure cell bacterial cultures. Libraries production was conducted with the DNA Prep Library Preparation Kit (Illumina, CA, USA). Whole DNA sequencing was performed using the Illumina iSeq 100 platform (Illumina, San Diego, CA, USA) with the iSeq Reagent Kit v.2, and 2 × 150 paired-end reads were generated. Read sets quality was evaluated using the FastQC v12.1 software (https://www.bioinformatics.babraham.ac.uk/projects/fastqc/, accessed on 1 January 2023). Illumina adapter sequences were recognized and removed using Trimmomatic v0.39 (https://github.com/usadellab/Trimmomatic, accessed on 1 January 2023). SPAdes v3.15.4 (https://github.com/ablab/spades, accessed on 1 January 2023) was used for de novo genome assembly based on Illumina short reads. The sequence type of the isolates was determined by scanning each assembly against the typing schemes obtained from the PubMLST website (https://pubmlst.org/, accessed on 1 January 2023) using MLST v2.11 (https://github.com/tseemann/mlst, accessed on 1 January 2023). The coding sequences of each strain were predicted by genome annotation using the RASTtk v2.0 software (https://github.com/TheSEED/RASTtk-Distribution, accessed on 1 January 2023). Genome assembly and annotation statistics were obtained using custom Python scripts based on the Biopython v1.79 package (https://biopython.org/, accessed on 1 January 2023). Genes involved in antimicrobial resistance were detected with AMRFinderPlus v3.10.30 (https://github.com/ncbi/amr, accessed on 1 January 2023). In addition, genes encoding for porins known to be involved in antimicrobial resistance were further investigated by aligning each nucleotide sequence against the respective amino acid sequence retrieved from the NCBI database using the blastx command from Diamond v2.1.6.16 (https://github.com/bbuchfink/diamond, accessed on 1 January 2023). The analysed sequences were OmpK35 [WP_004141771.1] and OmpK36 [WP_002913005.1] for *K. pneumoniae*; OprD [WP_000910004] and CarO [ABC46545.1] for *A. baumannii*; and OprD [P32722.1] and OprF [P13794.1] for *P. aeruginosa*.

### 5.2. Synergy Testing

Synergy analyses were performed in triplicate using the E-test method according to the manufacturers’ instruction (bioMe’rieux, France) and as previously described [34]. Briefly, antibiotic combinations tested by E-test strips were placed on the Muller Hinton agar plates, crossed with a 90° angle at the intersection between relative MIC for each isolate and incubated at 37 °C for 24 h.

Antibiotic combinations were CFD/PIP-TAZ, CFD/IMI-REL, CFD/MER-VAB, CFD/CAZ-AVI, CFD/AMP-SULB, CFD/FOS, SULB/PIP-TAZ, SULB/IMI-REL, SULB/MER-VAB, SULB/CAZ-AVI and SULB/FOS. The fractional inhibitory concentration (FIC) index (FICI) was calculated both for E-test and checkerboard assays by using the following formula: FICI = FIC of agent A + FIC B, where FIC A is the MIC of the combination/MIC of drug A alone, and FIC B is the MIC of the combination/MIC of drug B alone. FICI results were interpreted with the following criteria, as previously described [35]: synergy, FICI ≤ 0.5; independent interaction, 0.5 > FICI ≤ 4; and antagonism, FICI ≥ 4 [36].

## Figures and Tables

**Table 1 antibiotics-12-00858-t001:** Genotypic characteristics of CR strains included in this study.

Isolates	ST ^a^	β-Lactamase	Multidrug Efflux Pumps	Major Porins
				OmpK35
**CRE 1**	512	KPC-66, TEM, SHV-11, OXA-10, OXA-181, CMY-16	*emrD*, *oqxA*, *oqxB*	OmpK35, truncated at aa 41; OmpK36, INS135GD
**CRE 2**	307	KPC-3, TEM-1, SHV-28,	*emrD*, *oqxA*, *oqxB19*	OmpK35, truncated at aa 229; OmpK36, truncated at aa 183
**CR-Ab 1**	2	ADC-73, OXA-23, OXA-66	*adeC*, *amvA*	carO, variant III, Y245F; oprD, wt
**CR-Ab 2**	2	ADC-73, TEM-1, OXA-23, OXA-66, ftsl	*adeC*, *amvA*	carO, variant III, Y245F; oprD, wt
**CR-Pa 1**	298	OXA-848, BEL, PDC-16	*mexA*, *mexE*, *mexX*	oprD, variant C2, G425A; oprF, wt
**CR-Pa 2**	235	OXA-2, OXA-488, PDC-35, PER-1	*mexA*, *mexE*, *mexX*	oprD, variant C2, stop codon at aa 64, G425A; oprF, wt

^a^ ST, Sequence Type.

**Table 2 antibiotics-12-00858-t002:** Phenotypic characteristics of CR strains included in this study. Reduced susceptibility to antimicrobial antimicrobials is indicated in bold. ^a^: TAZ concentration fixed at 4 mg/L; ^b^: REL concentration fixed at 4 mg/L; ^c^: VAB concentration fixed at 8 mg/L; ^d^: AVI concentration fixed at 4 mg/L; ^e^: SULB concentration fixed at 4 mg/L.

Isolates	MIC (mg/L) ^1^							
	CFD ^2^	SULB ^3^	PIP-TAZ ^4^	IMI-REL ^5^	MER-VAB ^6^	CAZ-AVI ^7^	AMP-SULB ^8^	FOS ^9^
**CRE 1**	**16**	**>256**	**>256** ^a^	**4** ^b^	**16** ^c^	**48** ^d^	**>256** ^e^	**>256**
**CRE 2**	0.032	**>256**	**>256** ^a^	0.25 ^b^	4 ^c^	3 ^d^	**>256** ^e^	8
**CR-Ab 1**	**>256**	**>256**	**>256** ^a^	**>32** ^b^	**>256** ^c^	**>256** ^d^	**>256** ^e^	32
**CR-Ab 2**	0.125	**64**	**>256** ^a^	**>32** ^b^	**>256** ^c^	**48** ^d^	**>256** ^e^	**>256**
**CR-Pa 1**	0.5	**>256**	**12** ^a^	2 ^b^	**16** ^c^	**24** ^d^	**>256** ^e^	**>256**
**CR-Pa 2**	0.125	**>256**	8 ^a^	**>32** ^b^	**32** ^c^	8 ^d^	**>256** ^e^	**>256**

^1^ Applying EUCAST breakpoint; ^2^ CFD, cefiderocol; ^3^ SULB, sulbactam; ^4^ PIP-TAZ, piperacillin-tazobactam; ^5^ IMI-REL, imipenem-relebactam; ^6^ MER-VAB, meropenem-vaborbactam; ^7^ CAZ-AVI, ceftazidime-avibactam; ^8^ AMP-SULB, ampicillin-sulbactam; ^9^ FOS, fosfomycin.

**Table 3 antibiotics-12-00858-t003:** FICI values obtained from CFD in combination with PIP-TAZ, FOS, CAZ-AVI, IMI-REL, MER-VAB and AMP-SULB against CR strains included in this study. FICI values in synergistic range are reported in bold.

Isolates	CFD/PIP-TAZ ^1^	CFD/FOS ^2^	CFD/CAZ-AVI ^3^	CFD/IMI-REL ^4^	CFD/MER-VAB ^5^	CFD/AMP-SULB ^6^
**CRE 1**	1.25	**0.50**	**0.38**	0.63	0.63	0.88
**CRE 2**	1.00	0.86	0.83	1.00	0.75	1.47
**CR-Ab 1**	2.00	**0.44**	2.00	2.00	2.00	2.00
**CR-Ab 2**	1.50	1.01	1.75	2.00	2.00	1.50
**CR-Pa 1**	1.00	1.00	**0.50**	0.63	1.25	**0.50**
**CR-Pa 2**	2	1.75	2	0.63	1.26	2

^1^ CFD/PIP-TAZ, cefiderocol/piperacillin-tazobactam; ^2^ CFD/FOS, cefiderocol/fosfomycin; ^3^ CFD/CAZ-AVI, cefiderocol/ceftadidime-avibactam; ^4^ CFD/IMI-REL, cefiderocol/imipenem-relebactam; ^5^ CFD/MER-VAB, cefiderocol/meropenem-vaborbactam; ^6^ CFD/AMP-SULB, cefiderocol/ampicillin-sulbactam.

**Table 4 antibiotics-12-00858-t004:** FICI values obtained from PIP-TAZ, FOS, CAZ-AVI, IMI-REL or MER-VAB in combination with SULB against CR strains included in this study. FICI values in synergistic range are reported in bold.

Isolates	PIP-TAZ/ SULB ^1^	FOS/ SULB ^2^	CAZ-AVI/ SULB ^3^	IMI-REL/ SULB ^4^	MER-VAB/ SULB ^5^
**CRE 1**	2.00	0.75	**0.35**	1.50	1.25
**CRE 2**	2.00	1.25	0.71	1.25	0.56
**CR-Ab 1**	2.00	2.25	2.00	2.00	1.00
**CR-Ab 2**	2.00	1.00	0.83	1.75	0.88
**CR-Pa 1**	1.00	2.00	1.17	2.00	1.75
**CR-Pa 2**	1.00	2.00	0.88	2.00	2.00

^1^ PIP-TAZ/SULB, piperacillin-tazobactam/sulbactam; ^2^ FOS/SULB, fosfomycin/sulbactam; ^3^ CAZ-AVI/SULB, ceftadidime-avibactam/sulbactam; ^4^ IMI-REL/SULB, imipenem-relebactam/sulbactam; ^5^ MER-VAB/SULB, meropenem-vaborbactam/sulbactam.

## Data Availability

Not applicable.

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
