# Peer review of "Synergistic Activity of Cefiderocol in Combination with Piperacillin-Tazobactam, Fosfomycin, Ampicillin-Sulbactam, Imipenem-Relebactam and Ceftazidime-Avibactam against Carbapenem-Resistant Gram-Negative Bacteria"

_antibiotics, 2023, doi:10.3390/antibiotics12050858_

Round 1
Reviewer 1 Report
1. I suggest using AMR (antimicrobial resistance), not ‘antibiotic resistance’.
2. Line 9-10. ‘Limited therapeutic options’ are not the only or main reason for AMR development. Add ‘Limited therapeutic options is one of the reasons of…’.
3. Line 21. Decipher the abbreviations CRE and CR Ab.
4. Imipenem relebactam (IMI REL) is not mentioned in the title, unlike other antibiotics. Name it in the title, or modify the title in another way. I suggest a title: “Synergistic activity of Cefiderocol with some antibiotics against Carbapenem-resistant Gram-negative bacteria”
5. Key word ‘Bacteria and bacterial infections’ do not reflect the manuscript, you do not describe bacterial infections. Change it to ‘Carbapenem-resistant’, ‘Gram-negative bacteria’, ‘Cefiderocol’, and so on.
6. Line 35 and 38-39. DTI repetition.
7. Line 61. Decipher the abbreviations CAZ-AVI, in opposite to line 95.
8. A common problem in the manuscript is that abbreviations are not deciphered when first mentioned. Check the entire manuscript.
9. I suggest transferring Table 1 to 2.1 section.
10. Table 3. Decipher the abbreviation ‘FICI’. You decipher ‘FIC’ in methods but not ‘FICI’.
11. I think the tables should be moved immediately after the paragraph where a specific table was mentioned.
12. Conclusions should be presented in a separate section.
13. In the discussion, it is worth describing other publications that describe the state of the problem of resistance to cefiderocol in the world. And also what approaches already exist to the treatment of cefiderocol-resistant microorganisms.
1. Line 74. ‘chatecol’ or ‘catechol’?
2. Line 93. ‘tampicillin’ vs ampicillin.
3. Check the entire manuscript for technical errors.
Author Response
Reviewer #1
1. I suggest using AMR (antimicrobial resistance), not ‘antibiotic resistance’.
Authors’ reply and amendments: We thanks reviewer for the suggestions. We modified the text as requested
2. Line 9-10. ‘Limited therapeutic options’ are not the only or main reason for AMR development. Add ‘Limited therapeutic options is one of the reasons of…’.
Authors’ reply and amendments: We modified the text as requested
3. Line 21. Decipher the abbreviations CRE and CR Ab.
Authors’ reply and amendments: The text was modified as requested
4. Imipenem relebactam (IMI REL) is not mentioned in the title, unlike other antibiotics. Name it in the title, or modify the title in another way. I suggest a title: “Synergistic activity of Cefiderocol with some antibiotics against Carbapenem-resistant Gram-negative bacteria”
Authors’ reply and amendments: The title was modified as requested
5. Key word ‘Bacteria and bacterial infections’ do not reflect the manuscript, you do not describe bacterial infections. Change it to ‘Carbapenem-resistant’, ‘Gram-negative bacteria’, ‘Cefiderocol’, and so on.
Authors’ reply and amendments: The keywords was modified as requested
6. Line 35 and 38-39. DTI repetition.
Authors’ reply and amendments: The text was modified as requested
7. Line 61. Decipher the abbreviations CAZ-AVI, in opposite to line 95.
Authors’ reply and amendments: The text was modified as requested
8. A common problem in the manuscript is that abbreviations are not deciphered when first mentioned. Check the entire manuscript.
Authors’ reply and amendments: The manuscript was fully checked
9. I suggest transferring Table 1 to 2.1 section.
Authors’ reply and amendments: The table was moved to the section 2.1 as requested
10. Table 3. Decipher the abbreviation ‘FICI’. You decipher ‘FIC’ in methods but not ‘FICI’.
Authors’ reply and amendments: The text was modified as requested
11. I think the tables should be moved immediately after the paragraph where a specific table was mentioned.
Authors’ reply and amendments: Tables 2, 3, 4 are already after the paragraph 2.2 were are mentioned
12. Conclusions should be presented in a separate section.
Authors’ reply and amendments: The section was added as requested
13. In the discussion, it is worth describing other publications that describe the state of the problem of resistance to cefiderocol in the world. And also what approaches already exist to the treatment of cefiderocol-resistant microorganisms.
Authors’ reply and amendments: At this moment the literature related to the Cefiderocol-resistance and in particular to the most efficacy treatments for infections due to cefiderocol-resistant microorganisms are limited. Therefore, we hypotise that our results, although presenting in vitro results not related to the real clinical outcome, could be considered as preliminary data that could be used for further clinical studies for the evaluation of infections due to cefiderocol-resistant MDR Gram negative bacteria.
Comments on the Quality of English Language
1. Line 74. ‘chatecol’ or ‘catechol’?
Authors’ reply and amendments: The term was modified as requested
2. Line 93. ‘tampicillin’ vs ampicillin.
Authors’ reply and amendments: The term was modified as requested
3. Check the entire manuscript for technical errors.
Authors’ reply and amendments: The term was modified as requested
Reviewer 2 Report
Comments to the Author
Palombo et al reported in vitro synergistic activity of Cefiderocol in combination with pipera-cillin-tazobactam, fosfomycin, ampicillin-sulbactam, and ceftazidime-avibactam against Carbapenem-resistant Gram-negative bacteria. Overall, the manuscript was well organized and their findings are desirable for clinicians and researchers around the world. However, there are some minor concerns about this study that warrant consideration prior to publication. Please see my comments below.
1. For tables from 1-4, there are superscripts used for footnotes. I would suggest to use numbers for words and use letters for numbers. For example, to use “1 PIP-TAZ/SULB, piperacillin-tazobactam/sulbactam” instead of “ aPIP-TAZ/SULB, piperacillin-tazobactam/sulbactam” and use 4b instead of 42 to avoid confusion.
2. The authors mentioned additive a couple of times, but there was no FIC values for additive mentioned. Moreover, authors used No interaction for 0.5> FICI ≤ 4, however, they used indifference for some of the results. I would recommend to be consistent.
3. Authors didn’t say if they used quality control stain (s) and drugs for MIC and FIC test.
4. Authors didn’t mention if they did triplicate or duplicate for MIC experiments and synergy test either.
Minor editing of English language required
Author Response
Reviewer #2
Palombo et al reported in vitro synergistic activity of Cefiderocol in combination with pipera-cillin-tazobactam, fosfomycin, ampicillin-sulbactam, and ceftazidime-avibactam against Carbapenem-resistant Gram-negative bacteria. Overall, the manuscript was well organized and their findings are desirable for clinicians and researchers around the world. However, there are some minor concerns about this study that warrant consideration prior to publication. Please see my comments below. 1. For tables from 1-4, there are superscripts used for footnotes. I would suggest to use numbers for words and use letters for numbers. For example, to use “1 PIP-TAZ/SULB, piperacillin-tazobactam/sulbactam” instead of “ aPIP-TAZ/SULB, piperacillin-tazobactam/sulbactam” and use 4b instead of 42 to avoid confusion.
Authors’ reply and amendments: We thanks reviewer for the suggestions. The table was modified following referee’s suggestions
2. The authors mentioned additive a couple of times, but there was no FIC values for additive mentioned. Moreover, authors used No interaction for 0.5> FICI ≤ 4, however, they used indifference for some of the results. I would recommend to be consistent.
Authors’ reply and amendments: The methods was modified as requested
3. Authors didn’t say if they used quality control stain (s) and drugs for MIC and FIC test.
Authors’ reply and amendments: For synergy testing, no quality control strains were used.
4. Authors didn’t mention if they did triplicate or duplicate for MIC experiments and synergy test either.
Authors’ reply and amendments: The experiments are performed in triplicate. A sentence describing the method was added to the text
Comments on the Quality of English Language
Minor editing of English language required
Authors’ reply and amendments: We text was completely checked for English as requested
Round 2
Reviewer 1 Report
All my comments were taken into account. Thank you.